# AUTOMATIC TRUNCATION POSITION SELECTION IN SINGULAR VALUE DECOMPOSITION FOR LARGE LANGUAGE MODELS

## ABSTRACT

Model decomposition in large language models has drawn much attention due to its superiority and good interpretability, where activation-aware singular value decomposition (SVD) can achieve competitive performance by mitigating reconstruction errors brought by outliers in activation. However, the performance of the state-of-the-art SVD-based LLM compression method is limited to the selection of truncation positions. No work meticulously examines the details of this problem theoretically and empirically tests its correlation with model performance. To fill the research gap, we propose an efficient method that can *automatically select truncation positions*, namely AutoTrunc. In our work, we first analyze the correlation between truncation positions and the model performance. Then, the model layer importance is modeled based on the correlation, followed by mathematical proof to illustrate how to reach and obtain the optimal truncation position configuration for different layer types. Extensive experiments are carried out to verify our presumption and evaluate our proposed method. Our proposed AutoTrunc outperforms the state-of-the-art SVD-based LLM compression method, with perplexity scores dropping by 24.65% and 38.63% at the compression ratio of 50% in LLaMA-2-7B and LLaMA-2-13B, respectively. The code will be released upon acceptance.

## 1 INTRODUCTION

The large language model has been proven to perform exceptionally in natural language processing and related areas (Zhao et al., 2023). Despite the remarkable performance brought by billions of model parameters (Kaplan et al., 2020), modern large language models (LLMs) have presented considerable challenges to inference and deployment. It is necessary to reduce memory footprint during the inference to facilitate LLM deployment and democratization. To achieve this goal, researchers have proposed various model compression techniques (Miao et al., 2023), where model decomposition has recently drawn much attention due to its good interpretability. Despite some methods can achieve competitive performance without post-training (Yu & Wu, 2023; Yuan et al., 2023), it is still challenging to determine the truncation position for each layer. Selecting the most appropriate truncation position plays a crucial part in model performance, where a configuration of poor quality can lead to drastic performance degradation.

Many efforts have been devoted to this field and distinct methods are tried to address this problem. The most naive one is to adopt uniform truncation positions for all layers (Wang et al., 2024), which ignores the discrepancy between distinct layers to pursue each component to get equally compressed. Some propose to search for the best configuration in an iterative or adaptive manner (Hsu et al., 2022; Yuan et al., 2023; Chavan et al., 2024). Still, the search process is time-consuming since it involves expensive actual evaluation on the real-world dataset. Other methods leverage prior knowledge provided by artificial metrics designed by experts to guide the determination of truncation positions (Yin et al., 2024). The works discussed above have drawbacks to different extents, making it hard to reach a graceful balance between overhead and model quality. To the best of our knowledge, there is no work dedicated to studying how to determine truncation positions for each layer.

**Contributions** To mitigate the research gap illustrated above, we conduct a comprehensive analysis of the truncation position selection problem in our work. First, we meticulously examined the correlation between reconstruction error and inference quality through both theoretical and empirical methods. Then, we formalized the problem by formally defining it and mathematically proved it to be an NP-hard problem. In order to obtain solutions of good quality at an acceptable time expense, we first facilitated model performance estimation with learning-based layer importance modeling and then we proposed a highly efficient method to search the truncation configuration that is estimated to have the best model performance. In the end, we carried out comprehensive and extensive experiments to evaluate our proposed method AutoTrunc. To summarize, our contributions are listed below.

- We facilitate model performance estimation with learning-based layer importance modeling. The resulting scores of layer importance can be used with the layer's reconstruction error to effectively discriminate the model performance of different truncation configurations.

- We formalize the truncation position selection problem by formally defining it, and prove its hardness by a reduction from the 0-1 Knapsack Problem.

- We propose AutoTrunc, an efficient method that can automatically select appropriate truncation positions with only theoretical calculation, where we can prove that the resulting configurations can reach the upper bound of the estimated performance.

- We conduct comprehensive and extensive experiments to evaluate AutoTrunc. The results demonstrate the superiority of our proposed method, where the perplexity drops by 24.65% and 38.63% under the compression ratio of 50% in LLaMA-2-7B and LLaMA-2-13B, respectively.

## 2 TOWARDS THEORETICAL ESTIMATION ON MODEL PERFORMANCE

In this section, we give the preliminaries regarding the state-of-the-art SVD-based LLM compression technique SVD-LLM (Wang et al., 2024) and analyze the correlation between the truncation position selection (TPS) and its resulting model performance with both theoretical analysis and empirical experiments. In the end, we formalize the TPS problem by formally defining it.

### 2.1 PRELIMINARIES

The vanilla SVD method only focuses on the compression of pre-trained weights, whose compress loss can be denoted as Equation (1). Existing research found it suffers from reconstruction errors brought by outliers in activation (Yuan et al., 2023). To address this issue, activation-aware model decomposition (Yuan et al., 2023; Yu & Wu, 2023) proposes to minimize the reconstruction error of the activation instead of the pre-trained weights, whose compression loss now shifts to Equation (2) from Equation (1).

$$L = \|\mathbf{W} - \mathbf{W}'\|_F, \tag{1}$$

$$L = \|\mathbf{W}\mathbf{X} - \mathbf{W}'\mathbf{X}\|_F, \tag{2}$$

where $\mathbf{W}$ is the pre-trained weight of a linear layer and $\mathbf{W}'$ is its approximation, and $\mathbf{X}$ is the input. $\|\cdot\|_F$ denotes the Frobenius norm.

To improve the computing efficiency, Wang et al. (2024) propose to perform data whitening on the activation through Cholesky decomposition to capture data distribution. The process is described as follows. Let $\mathbf{S}$ be the result of Cholesky decomposition of the collected gram matrix $\mathbf{X}\mathbf{X}^T$, it performs singular value decomposition on $\mathbf{W}\mathbf{S}$ instead of $\mathbf{W}$, where compression loss, *i.e.,* Equation (2), has a similar characteristic as the vanilla SVD (Eckart & Young, 1936), *i.e.,* its square equals to the square sum of the truncated singular values (Theorem 1).

**Theorem 1.** *Given an input $X$, a weight matrix $\mathbf{W}$ with its two dimensions $m$ and $n$ where $m \leq n$, and its singular value decomposition results from $\mathbf{U}\Sigma\mathbf{V}^T = \mathbf{W}$. Let $\mathbf{S}$ be the Cholesky decomposition of $\mathbf{X}\mathbf{X}^T$. The compression loss of truncating the smallest singular values is $L^2 = \|\mathbf{W}\mathbf{X} - \mathbf{W}'\mathbf{X}\|_F^2 = \|\sum_{i=m+1}^{k} \sigma_i \boldsymbol{u}_i \boldsymbol{v}_i^T \mathbf{S}^{-1} X\|_F^2 = \sum_{i=m+1}^{k} (\sigma_i)^2$ and such truncating leads to the lowest loss.*

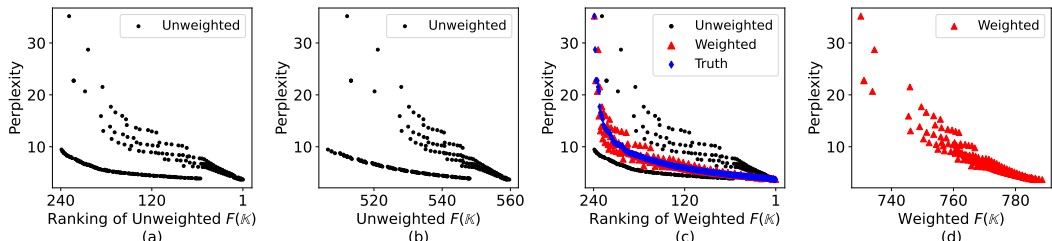

Figure 1: Unweighted (weighted) $F(\mathbb{K})$ and their rankings on Llama-2-70B, where data are collected through two different strategies under compression ratios ranging from 50% to 10%.

With the help of the closed-form solution of compression loss given in Theorem 1, we can define a performance score by measuring its relative error to assess the performance of a reconstructed linear layer $l$ with its truncation position $k_l$. The performance score of layer $l$ is defined as $f(k_l; l)$ in Equation (3).

$$f(k_l; l) = \frac{\sum_{i=1}^{k_l} \sigma_{l,i}^2}{\sum_{i=1}^{m_l} \sigma_{l,i}^2}, \tag{3}$$

where $\sigma_{l,i}$ denotes the $i$-th singular value in layer $l$, $m_l$ is the amount of singular values in layer $l$, $0 \leq f(k_l; l) \leq 1$, and $0 \leq k_l \leq m_l$. A high $f(k_l; l)$ indicates layer $l$ suffers few reconstruction errors.

## 2.2 CORRELATION BETWEEN PERFORMANCE SCORES AND MODEL QUALITY

Intuitively, with the given dataset, if every linear layer has a small compression loss, the compressed model will generally perform better. To this end, we treat the sum of all performance scores, *i.e.,* Equation (4), as a metric to theoretically estimate model performance after compression under a certain truncation position configuration $\mathbb{K}$.

$$F(\mathbb{K}) = \sum f(k_l; l), \text{ where } k_l \in \mathbb{K} \text{ and } l \in \mathbb{L} \tag{4}$$

Perplexity is a widely used metric to evaluate model performance, and it is closely related to the cross-entropy loss of the language model. The more likely it is for the language model to generate sentences that appear in the test set, the less the resulting perplexity is. The value of performance estimation $F(\mathbb{K})$ thus should strongly correlate with the perplexity. To end this, we test the correlation between the perplexity on Wikitext-2 (Merity et al., 2016) and $F(\mathbb{K})$, with data collected from two different strategies, the first of which is to compress all the layers uniformly, and the second is adopting greedy search to search the optimal truncation positions according to Equation (4). Both strategies are respectively applied to the model with all layers, w/o the first and last, w/o the first two and last two layers. Additionally, the compression ratio is gradually increased from 10% to 50%, making $6 \times 40 = 240$ data points available in total. The results are shown in Figure 1(a) and Figure 1(b).

Notably, the perplexity tends to decrease with the value $F(\mathbb{K})$ increasing overall. However, considering samples from different sources, there is a conspicuous divergence in the perplexity as $F(\mathbb{K})$ decreases, and it is therefore unreliable to estimate model performance based on the value of $F(\mathbb{K})$. One intuitive explanation for this phenomenon is the lack of layer importance, where in LLMs some layers are more important than others and the unweighted sum ignores this essential factor (Yin et al., 2024; Gromov et al., 2024; Men et al., 2024). We thus conjecture that for each layer, there is a factor representing its importance, which can make the weighted sum (also denoted as $F(\mathbb{K})$) of the performance scores and their importance factors able to estimate the model's quality. With effective layer importance factors, our defined $F(\mathbb{K})$ can be a powerful tool to efficiently estimate model performance, as illustrated in Figure 1(c) and Figure 1(d), where the divergence is significantly mitigated. The problem and solution regarding layer importance will be particularized in §3.

## 2.3 PROBLEM DEFINITION

Our vision of estimating the model performance by theoretical calculation can be divided into two smaller problems. The first problem is how to define and find a coefficient for each layer that can represent its importance so that their weighted sum (*i.e.,* the value of $F(\mathbb{K})$) can be used to discriminate the model performance. The second problem is determining the truncation position for each layer, where the value of $F(\mathbb{K})$ can reach maximal.

For the first problem, we notice the calculation of perplexity and the way we define performance scores have a strong relationship, intuitively. However, since the LLM is essentially a black-box model, it is impracticable to find layer importance where there is a strictly monotonic mapping between its $F(\mathbb{K})$ and the model's perplexity. Therefore, we try to establish a strong correlation between the value of $F(\mathbb{K})$ and the model performance. The first problem thus can be formalized as follows.

**Definition 1.** *Layer Importance Fitting problem (LIF problem). For a large language model, given its layers $l \in \mathbb{L}$, for each layer $l$, the layer importance fitting needs to find its coefficient $\alpha_l$, making their weighted sum with performance score, i.e., $F(\mathbb{K}) = \alpha_l f(k_l; l)$, where $k_l \in \mathbb{K}$, has a strong correlation with model's performance.*

With the given presumption (*i.e.,* it is solvable for LIF problem), where each layer has a coefficient that can represent its importance and their weighted sum can be leveraged to discriminate the model performance, we formalize the truncation selection problem in SVD-based LLM compression as follows.

**Definition 2.** *Truncation Position Selection problem (TPS problem). For a large language model, given its layers $l \in \mathbb{L}$ and their importance $\alpha_l$, layers' corresponding performance score function $f(k_l; l)$ under truncation position $k_l$, and memory consumption function $g(k_l; l)$, the objective of TPS is to determine the truncation position $k_l$ for each layer such that:*

$$\underset{\mathbb{K}}{argmax}\, F(\mathbb{K}) = \sum_{l \in \mathbb{L}} \alpha_l f(k_l; l), \text{ where } k_l \in \mathbb{K}$$

$$s.t. \sum_{l \in \mathbb{L}} g(k_l; l) \leq \mathcal{M} \tag{5}$$

$$f(k_l; l) \geq f_{min}^l$$

*where $\mathcal{M}$ represents the constraint on memory usage, and $f_{min}^l$ denotes the user-defined lower-bound constraint of layer $l$ to avoid excessive compression of certain layers.*

## 3 METHODOLOGY

For the LIF problem, even though there are many existing works whose proposed artificial metrics are proven to be effective in measuring layers' importance (Yin et al., 2024; Gromov et al., 2024; Men et al., 2024), it is infeasible to employ these metrics straightforwardly. This is because the correlation between $F(\mathbb{K})$ and the model performance is not captured through the priori knowledge. Adopting these artificial metrics will lead to the failure of performance estimation. Instead of employing the prior knowledge, we propose fitting layer importance with a learning-based method, where the resulting layer importance scores can successfully establish a correlation between $F(\mathbb{K})$ and the model performance.

### 3.1 LEARNING-BASED LAYER IMPORTANCE MODELING

One intuitive idea for solving the LIF problem is regression. However, predicting the perplexity is not our goal, and the linear regression can not fit highly complicated non-linear data, which significantly undermines its feasibility in solving the LIF problem. Moreover, simply employing linear regression can lead to negative values of the layer importance scores, which conflict with existing practices. Notably, our goal is to establish a correlation between $F(\mathbb{K})$ and the perplexity so that $F(\mathbb{K})$ can be used to guide us in selecting the best truncation position configuration for model decomposition. Therefore, we do not need to seek high accuracy on the perplexity prediction,

but rather, the accuracy of discriminating the model quality under different configurations. Consequently, we choose to solve the LIF problem with a ranking model. The ranking model uses $F(\mathbb{K})$ as its scoring function, whose parameters are essentially the layer importance. Once the ranking model is well trained, the $F(\mathbb{K})$ can strongly correlate with the perplexity. To this end, we introduce a listwise ranking method called LambdaRank (Burges et al., 2006) to learn to rank different truncation position configurations.

LambdaRank is a listwise learning-to-rank method that can capture information existing in the change of scores on metrics such as NDCG (Järvelin & Kekäläinen, 2002). Considering an ordered pair $(i, j)$, where $i$ has a higher relevant score, the loss function for the LambdaRank model can be formulated as Equation (6).

$$L_{ij} = \log(1 + \exp(s_i - s_j)) \cdot |\Delta Z_{ij}| \tag{6}$$

where $s_i$ is the score of item $i$ given by the ranking model and $|\Delta Z_{ij}|$ is the absolute value of the change value of a certain metric (*e.g.,* NDCG) if the two item's position is swapped.

In our scenario, the ranking model is just a single-layer perceptron without an activation function, *i.e.,* $F(\mathbb{K}) = \sum \alpha_l f(k_l; l)$, where $l \in \mathbb{K}$ and $\alpha_l > 0$. The data used to train the ranking model are collected following the same routine as described in §2.2, *i.e.,* truncation configurations under uniform compression and greedy search-based compression and their corresponding perplexity. In each epoch, we randomly generate four sequences with a length of 60 out of a total of 240 pairs and use these four sequences to train our ranking model. To prevent the importance score of the layer from becoming extremely large, we clamp the $\alpha$ within a certain range, *e.g.,* $[0.1, 10]$. The ranking model is evaluated with NDCG@100 on our collected data. Additionally, we employ the early stop strategy to select the best parameters as the layer importance to solve the TPS problem later. The evaluation of the ranking model will be presented in §4.2.

### 3.2 OPTIMAL CONFIGURATION TOWARDS SUB-LAYERS IN LLMS

We can prove the TPS problem is an NP-hard problem by a reduction from the 0-1 Knapsack Problem (see Appendix §B). It means we cannot easily find the optimal truncation configuration for the whole model at an acceptable time expense because of its vast solution space. However, when focusing on a single type of layers in Transformer-based (Vaswani et al., 2017) LLMs, the optimization challenge ceases to be NP-hard. This change in complexity is due to the uniformity in the dimensions of their weight matrices, defined as $m$ and $n$ where $m < n$, making it practical to pinpoint the optimal solution for different sub-layer types. In this context, we provide the upper bound of performance scores for specific types of layers and particularize the method for obtaining the corresponding solution.

**Lemma 1.** *The upper bound of the performance score $F(\mathbb{K}_s)$ for a specific type $s$ of layers $\mathbb{L}_s \subseteq \mathbb{L}$ in a large language model under memory usage constraint $\mathcal{M}_s$ is given by:*

$$F(\mathbb{K}_s) \leq \sum_{i=1}^{\lfloor \mathcal{M}/\beta_s \rfloor} \mathbb{V}_s \tag{7}$$

*where $\beta_s = (m+n)$ signifies the memory parameter associated for layers of type $s$, $\mathbb{V}_s = \{\gamma_l \sigma_{i,l}^2 | l \in \mathbb{L}_s\}$ represent the set of all values $\gamma_l \sigma_{i,l}^2$ for layers of type $s$, arranged in descending order.*

*Proof.* According to Equation (3) and Equation (5), we can derive the following:

$$F(\mathbb{K}_s) = \sum_{l \in \mathbb{L}_s} \alpha_l f(k_l; l) = \sum_{l \in \mathbb{L}_s} \frac{\alpha_l}{\sum_{i=1}^m \sigma_{i,l}^2} \sum_{i=1}^{k_l} \sigma_{i,l}^2.$$

Let $\gamma_l = \alpha_l / \sum_{i=1}^m \sigma_{i,l}^2$, then:

$$F(\mathbb{K}_s) = \sum_{l \in \mathbb{L}_s} \sum_{i=1}^{k_l} \gamma_l \sigma_{i,l}^2. \tag{8}$$

Since we focus on a specific type of layers, all of which have weight matrices of identical dimensions $m$ and $n$ (where $m < n$), we can determine the memory usage of layer $l$ at its truncation point $k_l$

using the formula $g(k_l; l) = k_l(m + n)$. Defining $\beta = m + n$ and given the memory constraint $\mathcal{M}_s$, we arrive at the following conclusion:

$$\sum_{l \in \mathbb{L}_s} g(k_l; l) = \sum_{l \in \mathbb{L}_s} \beta k_l \leq \mathcal{M}_s \Rightarrow \sum_{l \in \mathbb{L}_s} k_l \leq \frac{\mathcal{M}_s}{\beta}. \tag{9}$$

By integrating Equation (8) with Equation (9), we deduce that, given the memory usage limit $\mathcal{M}_s$, no more than $\mathcal{M}_s/\beta$ singular values can be selected. Let $\mathbb{V}_s = \{\gamma_l \sigma_{i,l}^2 | l \in \mathbb{L}_s\}$ represent the set of all values $\gamma_l \sigma_{i,l}^2$ for layers of type $s$, arranged in descending order. Consequently, the maximum value of $F(\mathbb{K}_s)$ is the sum of the top $\lfloor \mathcal{M}_s/\beta \rfloor$ values in the set $\mathbb{V}_s$. □

To prevent a certain layer from being excessively compressed, there is a user-defined lower bound of $f(k_l; l)$ where $f(k_l; l) \geq f_{\min}^l$. From which we can obtain $k_l^{\min}$ for each sub-layer $l \in \mathbb{L}_s$ where $f(k_l^{\min}, l) \geq f_{\min}^l$ since we already know every singular values. Once $k_l^{\min}$ values have been selected for each layer $l \in \mathbb{L}_s$, meeting the user-defined lower bound for $f(k_l; l)$, the focus shifts to satisfying the memory usage constraint. This scenario aligns with the problem described in Lemma 1, allowing us to easily derive the following corollary.

**Corollary 1.** *The upper bound of the performance score $F(\mathbb{K}_s)$ for a specific type $s$ of layers $\mathbb{L}_s \subseteq \mathbb{L}$ in a large language model under Equation (5) is given by:*

$$F(\mathbb{K}_s) \leq F_{min} + \sum_{i=1}^{k^{left}} \mathbb{V}_s^{left} \tag{10}$$

*where $F_{min} = \sum_{l \in \mathbb{L}_s} \sum_{i=1}^{k_l^{min}} \gamma_l \sigma_{i,l}^2$ represents the minimum performance score, set by the user-defined lower limit of $f(k_l; l)$ where $f(k_l; l) \geq f_{min}^l$. The term $k^{left} = \lfloor \mathcal{M}s/\beta \rfloor - \sum l \in \mathbb{L}_s$ represents the number of selections left, and $\mathbb{V}_s^{left}$ is the set of yet unselected values, sorted in descending order, after choosing $k_l^{min}$ values for each layer.*

Based on Corollary 1, identifying the solution that corresponds to the maximum performance score is straightforward. Initially, we choose $k_l^{\min}$ values for each layer $l \in \mathbb{L}_s$ to satisfy the user-defined lower limit of $f(k_l; l)$. Then, we iteratively select the largest value $\gamma_l \sigma_{i,l}^2$ until the memory usage surpasses the constraint $\mathcal{M}$. Upon selecting a value $\gamma_l \sigma_{i,l}^2$, the truncation position $k_l$ for the corresponding layer $l$ is incremented by 1. Consequently, this method of selecting truncation positions enables us to achieve the upper limit of $F(\mathbb{K}_s)$. The whole process is described in Algorithm 1.

Although we can obtain the optimal configuration for each layer type, it cannot guarantee that it is the optimal configuration for the whole model. To this end, we allocate the memory consumption budget to different

---

**Algorithm 1** Pseudocode for reaching the upper limit of $F(\mathbb{K}_s)$

---

**Input:** $f_{\min}^l, \gamma_l, \sigma_l^2$ where $l \in \mathbb{L}_s$
**Input:** Truncation budget $B = \lfloor \mathcal{M}s/\beta \rfloor$
1: **for** $l \in \mathbb{L}$ **do**
2:     $t^l \leftarrow f_{\min}^l$ // Initialize layer state, $t^l$ is truncation position of layer $l$
3:     $B \leftarrow B - t^l$ // Initialize budget state
4: **end for**
5: **repeat**
6:     $l \leftarrow$ Select the layer that has the largest $\gamma_l \sigma_{t^l+1,l}^2$
7:     $t^l \leftarrow t^l + 1$
8:     $B \leftarrow B - 1$
9: **until** $B = 0$

---

types of layers and find the optimal configuration for each layer type. Then, we traverse different ratios of budget allocation to try to get the best truncation configuration overall, where the calculation in different budget allocations can be conducted in a parallel manner for less time expense.

## 4 EXPERIMENTS

In this section, we carry out extensive experiments to evaluate our proposed method. First, we test the commonsense reasoning and generation performance of the compressed model under different compression ratios (§4.1). Secondly, we evaluate the effectiveness of our ranking model under different compression ratios (§4.2). In the end, we conduct an in-depth analysis of module sensitivity and budget allocation, and explore how the setting of $f_{\min}^l$ affects the model performance (§4.3).

Table 1: Zero-shot performance of top@1 accuracy on downstream task for compressed LLaMA-2-7B/13B/70B models, where the score in **Bold** indicates the best result at the same compression ratio

| Methods | Ratio | BoolQ | PIQA | WinoGrande | HellaSwag | ARC-E | ARC-C | OBQA | Avg. |
|---|---|---|---|---|---|---|---|---|---|
| Dense-7B | 0% | 0.7777 | 0.7905 | 0.6938 | 0.7592 | 0.7449 | 0.4625 | 0.442 | 0.6672 |
| SliceGPT | 20% | 0.3792 | 0.6126 | 0.5983 | 0.4428 | 0.4609 | 0.2841 | 0.306 | 0.4406 |
| SVD-LLM | | 0.5468 | 0.6513 | **0.6243** | 0.5173 | 0.4722 | 0.2782 | 0.380 | 0.4957 |
| Ours | | **0.6217** | **0.6839** | 0.6212 | **0.5492** | **0.5665** | **0.2944** | **0.386** | **0.5318** |
| SliceGPT | 30% | 0.3783 | 0.5555 | 0.5446 | 0.3517 | 0.3906 | 0.2457 | 0.280 | 0.3923 |
| SVD-LLM | | 0.5180 | 0.6001 | **0.5825** | 0.4185 | 0.4331 | 0.2543 | 0.340 | 0.4495 |
| Ours | | **0.6031** | **0.6170** | 0.5754 | **0.4392** | **0.4402** | **0.2602** | **0.352** | **0.4696** |
| Dense-13B | 0% | 0.8055 | 0.8041 | 0.7253 | 0.7941 | 0.7739 | 0.4915 | 0.456 | 0.6929 |
| SliceGPT | 20% | 0.3786 | 0.6224 | 0.6354 | 0.4730 | 0.4659 | 0.3191 | 0.386 | 0.4686 |
| SVD-LLM | | 0.7217 | 0.716 | **0.6843** | 0.5991 | 0.6212 | 0.3669 | 0.404 | 0.5876 |
| Ours | | **0.7422** | **0.7203** | 0.6827 | **0.6153** | **0.6305** | **0.3746** | **0.406** | **0.5959** |
| SliceGPT | 30% | 0.3783 | 0.5675 | 0.5770 | 0.3827 | 0.4087 | 0.2619 | 0.316 | 0.4132 |
| SVD-LLM | | 0.6401 | 0.6556 | 0.6393 | 0.4800 | 0.5059 | **0.3003** | 0.376 | 0.5139 |
| Ours | | **0.6606** | **0.6708** | **0.6440** | **0.5122** | **0.5156** | 0.2978 | **0.392** | **0.5276** |
| **Dense-70B** | 0% | 0.8388 | 0.8275 | 0.7782 | 0.838 | 0.8072 | 0.5717 | 0.486 | 0.7353 |
| SliceGPT | 20% | 0.4394 | 0.6801 | 0.7214 | 0.5716 | 0.6864 | 0.4394 | 0.436 | 0.5678 |
| SVD-LLM | | 0.6422 | 0.7824 | **0.7664** | 0.7629 | **0.7912** | 0.5410 | 0.450 | 0.6766 |
| Ours | | **0.6972** | **0.7960** | 0.7545 | **0.7760** | 0.7883 | **0.5461** | **0.454** | **0.6875** |
| SliceGPT | 30% | 0.3783 | 0.6235 | 0.6701 | 0.4491 | 0.5404 | 0.3285 | 0.392 | 0.4831 |
| SVD-LLM | | 0.6235 | 0.7448 | 0.7427 | 0.6735 | 0.7449 | 0.4957 | 0.420 | 0.6350 |
| Ours | | **0.6306** | **0.7688** | **0.7561** | **0.7323** | **0.7590** | **0.4991** | **0.440** | **0.6551** |

**Baselines**    We compare our proposed method with the state-of-the-art model decomposition method SVD-LLM (Wang et al., 2024) and a structured pruning method SliceGPT (Ashkboos et al., 2024) in commonsense reasoning tasks. Additionally, we also add ASVD (Yuan et al., 2023) and an unstructured pruning method LLMPruner (Ma et al., 2023) as the baseline in the generation task.

**Models and Datasets**    The model we adopted are from LLaMA-2 family (Touvron et al., 2023) (LLaMA-2-7B, LLaMA-2-13B, LLaMA-2-70B). For a fair and reliable comparison, we evaluate our proposed method on seven widely adopted commonsense reasoning datasets in a zero-shot manner. Datasets are BoolQ (Clark et al., 2019), PIQA (Bisk et al., 2020), WinoGrande (Sakaguchi et al., 2019), HellaSwag (Zellers et al., 2019), ARC-easy/challenge (Clark et al., 2018), Open-BookQA (Mihaylov et al., 2018) from publicly available benchmark suite called Language Model Evaluation Harness framework (Gao et al., 2024). For the generation task, we adopted a common high quality dataset WikiText-2(Merity et al., 2016) to measure the model's perplexity.

**Implementation Details**    The process of model decomposition is kept the same as SVD-LLM. For the ranking model, we use AdamW (Loshchilov & Hutter, 2019) as our optimizer with parameters clamping between 0.1 and 10. The number of data pairs collected to train our ranking model is 240, and we apply the early stop strategy to select the ranking model with the best NDCG@100 score in 1000 iterations. As for the memory usage budget allocation, we traverse the ratios for MLP and Attention sublayer between 0.2 and 0.8 with 601 steps. Since there is no strict monotonic correlation between $F(\mathbb{K})$ and the perplexity, we evaluate the performance of truncation position configurations whose $F(\mathbb{K})$ is within the top-10. Additionally, to leverage NVIDIA hardware[1], we set the granularity of truncation to 16.

## 4.1 Overall Performance

**Commonsense Reasoning**    To evaluate the overall performance of our proposed method, we compared the zero-shot performance on the seven downstream commonsense reasoning datasets with top@1 accuracy, where the foundation models are compressed to different degrees and the results are

---

[1]https://docs.nvidia.com/cuda/cublas/index.html#tensor-core-usage

Table 2: Perplexity($\downarrow$) of compressed methods for LLaMA-2 family on WikiText-2.

| Method | Ratio | LLaMA-2 | | |
|---|---|---|---|---|
| | | 7B | 13B | 70B |
| Dense | 0% | 5.11 | 4.57 | 3.12 |
| LLM-Pruner | | 10.55 | 9.67 | - |
| SliceGPT | | 9.70 | 8.21 | 5.76 |
| ASVD | 20% | 9.38 | 6.33 | - |
| SVD-LLM | | 8.07 | 6.18 | 4.34 |
| AutoTrunc (Ours) | | **7.80** | **6.01** | **4.23** |
| LLM-Pruner | | 18.25 | 17.59 | - |
| SliceGPT | | 15.42 | 12.68 | 8.09 |
| ASVD | 30% | 364.53 | 20.77 | - |
| SVD-LLM | | 11.40 | 7.93 | 5.07 |
| AutoTrunc (Ours) | | **10.72** | **7.42** | **5.00** |

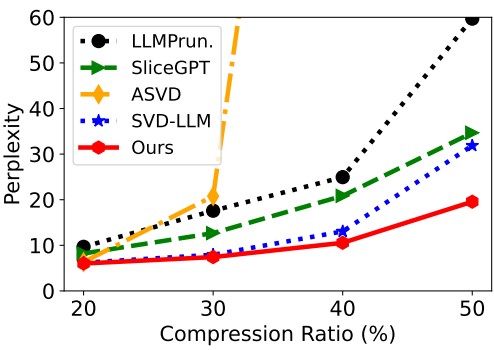

Figure 2: Perplexity($\downarrow$) on WikiText2 under different compression ratios. on LLaMA-2-13B.

shown in Table 1. As shown, AutoTrunc consistently outperforms SVD-LLM as well as the structured pruning method SliceGPT in multiple downstream datasets. For LLaMA-2-7B, AutoTrunc outperforms SVD-LLM 7.23% and 4.47% under 20% and 30% compression ratios, respectively.

**Generation Quality** We tested models' perplexity scores under different compression ratios on WikiText-2 to evaluate generation quality. The results are reported in Table 2. We also add an unstructured pruning method called LLM-Pruner (Ma et al., 2023) as an additional baseline on LLaMA-2-7B/13B. Our AutoTrunc outperforms SVD-LLM under the compression ratios of 20% and 30% in different LLaMA-2 models. Furthermore, we increase the compression ratio up to 50% for LLaMA-2-7B/13B and calculate the perplexity to see how our method performs under a high compression ratio. The perplexity variation on LLaMA-2-13B is shown in Figure 2. When the compression ratio is 50%, our AutoTrunc has the lowest perplexity, with 41.34 (vs. 54.86 by SVD-LLM) on LLaMA-2-7B and 19.56 (vs. 31.87 by SVD-LLM) on LLaMA-2-13B. The perplexity drops 24.65% and 38.63%, respectively.

### 4.2 PERFORMANCE OF RANKING MODEL

To verify our presumption (*i.e.,* the LIF problem) and evaluate our ranking model, we traversed all feasible solutions to explore the correlation between $F(\mathbb{K})$ and perplexity. For the solution space, we tried every possible budget allocation ratio between 20% and 80% at a step length of 0.1%. The results are shown in Figure 3. Notably, Figure 3(a) and Figure 3(c) indicate a strong correlation between $F(\mathbb{K})$ and perplexity, where they share a similar pattern of variation when the budget allocation ratio varies. As $F(\mathbb{K})$ increases, the resulting perplexity decreases until $F(\mathbb{K})$ reaches its maximum and then declines where the perplexity drops to its minimum before deteriorating again.

We ranked all the results according to their scores in $F(\mathbb{K})$ in descending order and tested their perplexity, where the results are shown in Figure 3(b) and Figure 3(d). Although, theoretically, there is no strictly monotonic correlation between $F(\mathbb{K})$ and the perplexity, it is clear that our learned layer importance scores are proved to be effective where $F(\mathbb{K})$ can discriminate the performance of models after decomposition. To evaluate effectiveness with quantitative metrics,

Table 3: NDCG($\uparrow$) of the ordered list predicted by our method under different compression ratios on LLaMA-2-7B

| Ratio | 20% | 30% | 40% | 50% |
|---|---|---|---|---|
| NDCG@10 | 0.915 | 0.960 | 0.999 | 0.996 |
| NDCG@20 | 0.915 | 0.957 | 0.999 | 0.995 |
| NDCG@30 | 0.916 | 0.959 | 0.998 | 0.997 |

we calculate the NDCG score of the ordered list predicted by our method and the results are reported in Table 3, where high NDCG scores indicate our method can easily find those configurations with similarly low perplexity scores.

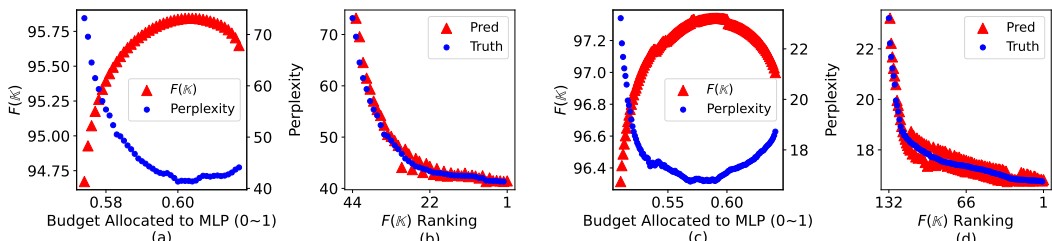

Figure 3: The variation and correlation between values of $F(\mathbb{K})$ and perplexity($\downarrow$) on LLaMA-2-7B, where (a) and (b) are for the 50% compression ratio, (c) and (d) are for the 40%.

## 4.3 IN-DEPTH ANALYSIS

**Layer Sensitivity** Our proposed method finds truncation position configurations for different layers following the guidance presented in §3.2. During the process, AutoTrunc iteratively increases truncation positions that lead to the maximal increment of $F(\mathbb{K})$. To explore the layer's sensitivity, we tried different $f_{\min}^l$ values from 0.85 to 0.95 to see the variation of $k_{\min}^l$, where $k$ is normalized to $[0, 1]$ and 1 denotes the maximal profitable truncation position. The results are visualized in Figure 4. Notably, it is clear that there is a significant difference between different modules. For instance, "q_proj" and "k_proj" have much smaller $k_{\min}^l$ compared with other modules even under a high $f_{\min}^l$ value, and the rapid change on $k_{\min}^l$ in their deeper layers suggests these layers less sensitive than their shallow layers. A similar phenomenon can be witnessed in the shallow layers of "gate_proj". As for most layers in "v_proj", "o_proj", and "down_proj", a little change of $k_{\min}^l$ under a certain $\Delta f_{\min}^l$ suggests their sensitivity to alteration in the truncation position. Sensitive modules and layers are more likely to consume the budget since they can bring maximal increment to $F(\mathbb{K})$.

**Budget Allocation** To verify our analysis, we tested how the budget is allocated to different layers and their corresponding contribution to $\Delta F(\mathbb{K})$. Additionally, we also recorded how different layers make up the final proportion of $F(\mathbb{K})$ and parameters. The results are reported in Table 4, where we can notice that about 75% budget is allocated to "q_proj", "k_proj", and "gate_proj". About 85% contribution to $\Delta F(\mathbb{K})$ is attributed to these three layers, ending up with the highest three proportions of $F(\mathbb{K})$. The resulting performance (PPL 41.34 vs. 54.86 by SVD-LLM) demonstrates the effectiveness of budget allocation, indicating it can automatically find the appropriate truncation positions. Besides, we also tested model performance on commonsense reasoning tasks under high compression ratios to demonstrate the superiority of AutoTrunc, where the results are reported in Table 5.

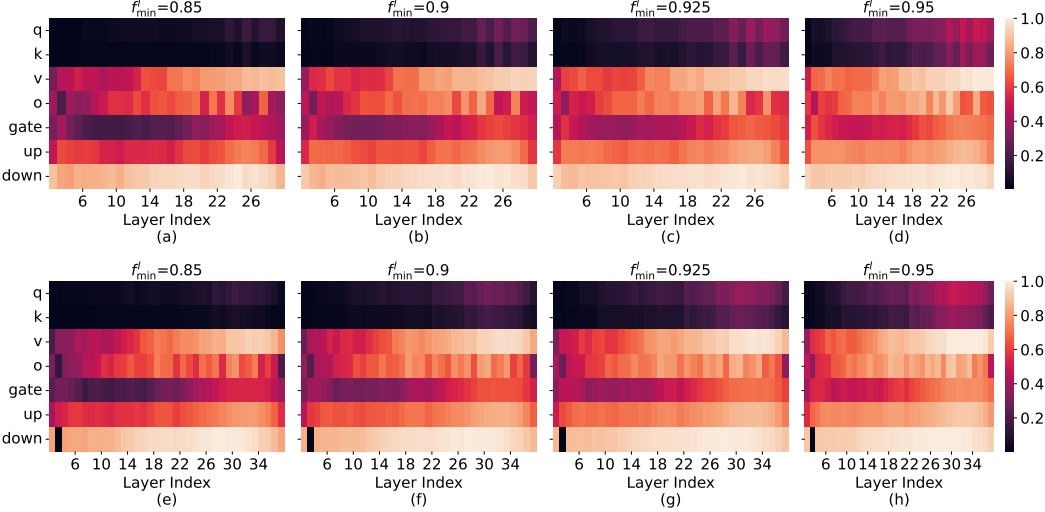

Figure 4: $k_{\min}^l$ under different $f_{\min}^l$ settings in LLaMA-2-7B (a-d) and LLaMA-2-13B (e-h). Given a fixed $\Delta f_{\min}^l$, a drastic change in $k_{\min}^l$ indicates that the layer $l$ is sensitive to compression.

Table 4: Proportion of budget, $F(\mathbb{K})$, and model parameters in LLaMA-2-7B under 50% compression ratio.

| Layers | $\Delta F(\mathbb{K})$ | Budget | $F(\mathbb{K})$ | Params (Init) | Params |
|---|---|---|---|---|---|
| q | 0.267 | 0.190 | 0.188 | 0.030 | 0.065 |
| k | 0.309 | 0.190 | 0.197 | 0.062 | 0.062 |
| v | 0.054 | 0.089 | 0.085 | 0.099 | 0.097 |
| o | 0.041 | 0.086 | 0.092 | 0.085 | 0.085 |
| gate | 0.280 | 0.361 | 0.184 | 0.168 | 0.209 |
| up | 0.023 | 0.054 | 0.122 | 0.251 | 0.208 |
| down | 0.026 | 0.031 | 0.133 | 0.341 | 0.274 |

Table 5: Comparison of commonsense reasoning performance on LLaMA-2 models under different compression ratios and methods

| Models | Methods | Ratio | | |
|---|---|---|---|---|
| | | 30% | 40% | 50% |
| | ASVD | 0.373 | 0.352 | 0.347 |
| 7B | SVD-LLM | 0.450 | 0.387 | 0.362 |
| | AutoTrunc | **0.470** | **0.404** | **0.370** |
| | ASVD | 0.525 | 0.393 | 0.347 |
| 13B | SVD-LLM | 0.514 | 0.435 | 0.380 |
| | AutoTrunc | **0.528** | **0.460** | **0.392** |

**Impact of the user-defined lower bound $f^l_{\min}$** We conducted grid search on LLaMA-2-7B under 50% compression ratio to explore the impact of $f^l_{\min}$. Motivated by Gromov et al. (2024), we grouped the whole model into two parts: the first/last 16 layers, where the $f^l_{\min} \geq 0.8$, and shallow layers is not less than the deeper layers. The results are shown in Figure 5, where an appropriate setting on $f^l_{\min}$ can prevent some layers from being excessively compressed, resulting in a better performance. Subtle control on $f^l_{\min}$ will be deferred to future works.

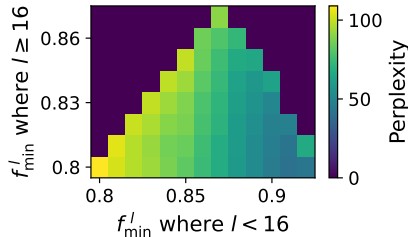

Figure 5: Variation of the perplexity($\downarrow$) on LLaMA-2-7B under 50% compression ratios with different $f^l_{\min}$ settings.

## 5 RELATED WORKS

**Large Language Model Compression** LLM compression has drawn much attention for its crucial part in LLM deployment. According to the granularity and methodology, they can be roughly categorized into the following types: unstructured-pruning (Ma et al., 2023; Yin et al., 2024; Dong et al., 2024); Quantization (Frantar et al.); structured-pruning (Ashkboos et al., 2024; Men et al., 2024); knowledge distillation (Du et al., 2024); model decomposition (Yuan et al., 2023; Wang et al., 2024; Yu & Wu, 2023). Different types of LLM compression techniques have their own strength and drawbacks. They are orthogonal and can be applied at the same time. There is no particular technique that can significantly outperform others in terms of overhead, efficiency, generation quality, and performance on the downstream tasks at the same time (Miao et al., 2023).

**Model Decomposition for LLMs** Vanilla SVD suffers from reconstruction errors brought by outliers. To mitigate the error, researchers have proposed different methods to capture data distribution in the input and output. Yu & Wu (2023) noticed low-rank structure does not exist in the pretrained weights but their features, proposing to approximate the features ($\mathbf{WX}$) instead of the weight ($\mathbf{X}$) with Atomic Feature Mimicking (AFM). Activation-aware singular value decomposition (Yuan et al., 2023) and SVD-LLM (Wang et al., 2024) employ the same idea, using improved SVD to compress LLMs. To summarize, all advanced model decomposition methods realize the problem incurred by outliers and try to solve it, *i.e.,* AFM-based (Yu & Wu, 2023; Kaushal et al., 2023; Ji et al., 2024; Chavan et al., 2024), improved SVD (Yuan et al., 2023; Chavan et al., 2024), masking (Li et al., 2023), and fine-grained decomposition (Liu et al., 2024).

## 6 CONCLUSION

In this paper, we propose AutoTrunc, an efficient method to address the truncation position selection problem with only theoretical calculation. It facilitates model performance estimation with learning-based layer importance modeling, followed by searching the truncation configurations that are most likely to have the best model performance. Extensive experiments are carried out to evaluate our proposed method AutoTrunc. We have demonstrated the superiority of AutoTrunc under different compression ratios on 8 datasets and 3 models from the LLaMA-2 family. Compared with the state-of-the-art method, the perplexity on WikiText-2 by 24.65% and 38.63% in LLaMA-2-7B and LLaMA-2-13B, under 50% compression ratio drops, respectively.

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

APPENDIX

This appendix aims to provide additional information and in-depth analysis to supplement the main content, which mainly includes three parts: further discussion, hardness of truncation position selection problem, and supplementary materials. In the first section, we discuss the advantages and limitations of AutoTrunc, as well as our insight on the implications of AutoTrunc. Then, we provide details regarding the hardness of the truncation position selection problem in the second part, and more experiment results in the last section.

## A FURTHER DISCUSSION

### A.1 ADVANTAGES AND LIMITATIONS

The major advantage of AutoTrunc is it can select appropriate truncation positions for each layer automatically, where the whole process is highly efficient since it only requires theoretical calculation. However, the superiority of AutoTrunc largely depends on the performance of the ranking model, *i.e.,* $F(\mathbb{K})$ in the main content. A ranking model with high accuracy in discriminating truncation configurations helps improve the quality of the derived configuration.

### A.2 FEASIBILITY OF GENERALIZATION

Our proposed AutoTrunc can be generalized and integrated into other model decomposition-based LLM compression methods as long as they satisfy the following three conditions: (1) It can define a function $f(k_l; l)$ to estimate compressed layer $l$'s performance based on its truncation position $k_l$; (2) It collects enough data pairs that include values of $f(k_l; l)$ for each layer of the model and its resulting perplexity score. With these two key component, we can successfully build a metric that strongly correlated with the model performance.

## B HARDNESS OF TRUNCATION POSITION SELECTION PROBLEM (TPS)

We argue for the use of an approximate algorithm to address the problem of selecting truncation positions. We introduce the concept of the Truncation Position Verification Problem (TPV), which is a more constrained version of the broader Truncation Position Selection Problem. We demonstrate that even this limited version remains computationally challenging.

**Definition 3.** *Truncation Position Verification Problem (TPV). Given the layers $l \in \mathbb{L}$ of a large language model, we have performance parameter $\gamma_l$ and memory consumption parameter $\beta_l$. The objective of this problem is to find the truncation position $k_l$ for each layer such that:*

$$F(\mathbb{K}) = \sum_{l \in \mathbb{L}} (\gamma_l \sum_{i=1}^{k_l} \sigma_{i,l}^2) \geq \mathcal{F}, \text{ where } k_l \in \mathbb{K},$$
$$s.t. \sum_{l \in \mathbb{L}} \beta_l k_l \leq \mathcal{M}, \quad k_l \geq k_{min},$$

(11)

*where $\sigma_{l,i}$ denotes the $i$-th singular value in layer $l$, $\mathcal{M}$ represents the constraint on memory usage, $\mathcal{F}$ is the desired performance, and $k_{min}$ is the minimum threshold for truncation positions across all layers, ensuring that no layer is compressed too aggressively.*

**Theorem 2.** *The problem of TPS is NP-hard.*

*Proof.* We will prove the theorem by showing that even a simpler problem of verifying whether $F(\mathbb{K})$ is larger than a given value (*i.e.,*, TPV problem) is NP-complete, which is proved by a reduction from the 0-1 Knapsack Problem.

It is obvious that TPV problem belongs to NP. It simply requires calculating the performance function $F(\mathbb{K})$ and verifying if the performance meets the specified target, while also ensuring that the total memory usage and each truncation position satisfy the constraints. This can be done in polynomial time. Therefore, TPV problem belongs to NP.

To show that TPV problem is NP-complete, we will reduce it from the 0-1 Knapsack Problem. Consider an arbitrary instance of the 0-1 Knapsack Problem, which includes $n$ binary variables $\{x_1, x_2, ..., x_n\}$, where $w[j]$ and $p[j]$ represent the weight and profit of item $x_j$, respectively. The backpack can hold items up to a total weight of $W$. The objective is to find whether or not there exists a "solution" with profit no less than $P$, where $P$ is the desired profit. Then, we construct a TPV corresponding to the 0-1 Knapsack Problem instance as follows:

- The model is composed of $n$ layers, with each layer featuring two possible truncation positions: 0 or 1, meaning $k_l \in 0, 1$. The minimum truncation position, $k_{min}$, is set to 0.

- For the $l$th layer, $\beta_l = w[l]$. And if the truncation position $k_l = 1$, then $\gamma_l \sigma_{1,l}^2 = p[l]$.

- The constraint on memory usage is equivalent to the knapsack's capacity, and the target performance matches the desired profit, *i.e.,*, $\mathcal{M} = W$ and $\mathcal{F} = P$.

Hence, the TPV problem instance is defined as identifying the truncation position $k_l$ for each layer to satisfy the following conditions:

$$\sum_{l=1}^{n} (k_l p[l]) \geq P, \text{ where } k_l \in \{0, 1\},$$
$$s.t. \sum_{l \in \mathbb{L}} w[l] k_l \leq W. \tag{12}$$

Then, we demonstrate that a solution to the 0-1 Knapsack Problem instance exists if and only if a solution to the TPV problem instance exists. It is clear that if a solution for the 0-1 Knapsack Problem exists—where items are chosen (corresponding to selecting truncation positions $k_l$) to achieve a profit of at least $P$ without surpassing the knapsack's capacity $W$—then this selection approach also addresses the TPV problem instance as outlined in Equation (12).

Conversely, suppose there is a solution for the TPV problem. In that case, it can be adapted to solve the 0-1 Knapsack Problem by aligning the chosen truncation positions with the items selected for the knapsack. This alignment is possible because the solution to the TPV problem ensures a knapsack profit of at least $P$ while keeping the total weight under $W$. This implies that solving the Truncation Position Verification Problem (TPV) is at least as hard as solving the 0-1 Knapsack Problem, which is known to be NP-complete. □

## C SUPPLEMENTARY MATERIALS

### C.1 INFERENCE THROUGHPUT

We tested the throughput under different compression ratios and batch size on a single GPU of A800 and CPU of Intel Xeon 8358P. The sequence length has been fixed to 32, the number of generated tokens to 128, and the decoding strategy is greedy sampling. The results are reported in Table 6, where we did not compress KV cache, and the speedup is thus not significant as illustrated in SVD-LLM and ASVD.

Figure 6: Throughput (tokens/sec) under different compression ratios and batch size.

| Batch size | 0% | 20% | 40% | 60% |
|---|---|---|---|---|
| 256 | 2933 | 2989 | 3116 | 3277 |
| 128 | 2521 | 2568 | 2705 | 2862 |
| 64 | 1954 | 1959 | 2026 | 2118 |

### C.2 COMMONSENSE REASONING PERFORMANCE

We provide detailed experiment results regarding Table 5. As we already give the zero-shot performance results under 30% compression ratio in Table 1, we only reported results under 40% and 50% compression ratios in Table 6.

Table 6: Zero-shot performance of top@1 accuracy on downstream task for compressed LLaMA-2-7B/13B models, where the score in **Bold** indicates the best result at the same compression ratio

| Methods | Ratio | BoolQ | PIQA | WinoGrande | HellaSwag | ARC-E | ARC-C | OBQA | Avg. |
|---------|-------|-------|------|------------|-----------|-------|-------|------|------|
| Dense-7B | 0% | 0.7777 | 0.7905 | 0.6938 | 0.7592 | 0.7449 | 0.4625 | 0.442 | 0.6672 |
| ASVD |  | **0.4434** | 0.5016 | 0.4807 | 0.2554 | 0.2517 | **0.2773** | 0.254 | 0.3520 |
| SVD-LLM | 40% | 0.3786 | 0.5555 | 0.5478 | 0.3408 | 0.3620 | 0.2287 | 0.292 | 0.3865 |
| Ours |  | 0.3939 | **0.5680** | **0.5533** | **0.3632** | **0.3691** | 0.2517 | **0.326** | **0.4036** |
| ASVD |  | **0.3813** | 0.4995 | 0.4878 | 0.2569 | 0.2622 | **0.2722** | 0.270 | 0.3471 |
| SVD-LLM | 50% | 0.3783 | **0.5305** | 0.5225 | 0.2989 | 0.3064 | 0.2363 | 0.260 | 0.3618 |
| Ours |  | 0.3783 | 0.5288 | **0.5375** | **0.3022** | **0.3089** | 0.2312 | **0.300** | **0.3696** |
| Dense-13B | 0% | 0.8055 | 0.8041 | 0.7253 | 0.7941 | 0.7739 | 0.4915 | 0.456 | 0.6929 |
| ASVD |  | **0.5355** | 0.5566 | 0.5383 | 0.3067 | 0.3165 | 0.2261 | 0.274 | 0.3934 |
| SVD-LLM | 40% | 0.4119 | 0.5990 | 0.6046 | 0.3976 | 0.4087 | **0.2739** | 0.348 | 0.4348 |
| Ours |  | 0.5327 | **0.6094** | **0.6077** | **0.4192** | **0.4331** | 0.2722 | **0.348** | **0.4603** |
| ASVD |  | 0.3786 | 0.5120 | 0.4893 | 0.2623 | 0.2748 | **0.2526** | 0.260 | 0.3471 |
| SVD-LLM | 50% | 0.3783 | 0.5381 | 0.5391 | 0.3232 | 0.3401 | 0.2304 | 0.308 | 0.3796 |
| Ours |  | **0.3826** | **0.5501** | **0.562** | **0.3405** | **0.3527** | 0.2389 | **0.314** | **0.3915** |

## C.3 DETAILS OF EXPERIMENTS ON USER-DEFINED $f_{\text{MIN}}^l$

We here provide details of experiments regarding impacts of user-defined $f_{\min}^l$ in §4.3. The information related to the experiment settings and results are reported in Table 7. In this experiment, we noticed the select configurations are clustered around the ratio of 0.6, whose density of distribution is shown in Figure 7.

Table 7: Experiment details

| Argument | Values |
|----------|--------|
| Search Space[1] | [0.3, 0.7] |
| Steps[2] | 81 |
| #Feasible Configs | 1064 |
| #Selected Top-1 Config | 59 |
| The Best Perplexity | 41.33 |
| The Worset Perplexity | 109.08 |

[1] The ratio of budget allocated to MLP.
[2] The number of steps it needs to traverse the search space.

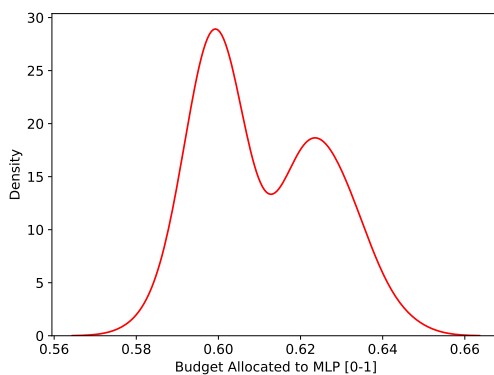

Figure 7: Distribution of budget allocation ratios in selected configurations

