# OpenReview forum: "Automatic Truncation Position Selection in Singular Value Decomposition for Large Language Models"
_ICLR.cc/2025/Conference — ICLR 2025 Conference Withdrawn Submission_

### Official Review · Reviewer_YQo2 · 2024-10-28

**Soundness:** 3
**Presentation:** 4
**Contribution:** 4
**Rating:** 6
**Confidence:** 4

**Summary:**

The paper addresses the critical problem of compressing large language models using SVD based decomposition. The significant contribution of the paper is a theoretical backing with empirical evidence to automatically truncate the singular values/vectors of each layer instead of applying a uniform low-rank on each layer like in the SVD-LLM method case. The paper has a learnable strategy that learns to decompose a given layer in an LLM using layer importance modeling.

**Strengths:**

- The paper is well-written and easy to follow along.
- The contributions in the paper are nicely positioned w.r.t the state-of-the-art in the literature.
- The strongest point(s) of the paper are i) applying SVD-LLM on each layer of the LLM in an adaptive manner ii) learning the layer importance, applying the lower bounds on the compression ratios at each layer (of course this is different from the over-all compression of the entire LLM), LambdaRank for listwise ranking  iii) empirically quantification on sub-layers the correlation between performance and the model quality.

**Weaknesses:**

Follow the questions section

**Questions:**

Following are a few questions that can benefit the paper
- Is it possible to quantify both with experiments and asymptotically the extra effort required to derive layer-wise ranks? This can help demonstrate the gains of the method relative to the SVD-LLM.
- What was the observation when the lower-bound was not applied and yet the AutoTrunc is allowed to automatically configure the layer-wise low-ranks for decomposition? It is understandable the performance might be taking a hit significantly, probably in many cases, worse than SVD-LLM, but it is worth showing that as a baseline to demonstrate the efficacy the lower-bound. This maybe especially useful to highlight the fact that ideal low-rank decomposition may not necessarily be good all the time and hence having a constrained (with the lower-bounds on each layer) adaptive/automatic truncation can be justified even better.
- In the similar spirit, is it a better strategy to set an overall compression ratio as a single hyper-parameter as opposed to set a low-bound on each layer and let the algorithm decide the truncation of the singular values in each layer? This in itself might be a new direction can be time-consuming for this review, but certainly a potential future direction and a new method.
- The proposed method is applied only on LLaMa-2-7/13/70B models, why not apply on other family of LLMs such as Mistral/Phi-3/X/Y/Z even if it is in the realm of 7B or lower parameter models? The questions here are, i) the learned `\alpha(s)` or even the strategy of AutoTrunc can be transferred to other family of models, ii) If the answer is yes, what is common/different in all these models that is actually contributing to the improved performances or a layer-wise study can be better explained, iii) if the answer is no, then can this be transferred easily to new family of LLMs or do we have to repeat the whole method from scratch for a new LLM architecture?
- The performance of the model drops >=50% compression ratio when compared to SVD-LLM, any hunches as to why?
- Is it possible to compare and contrast different SOTA methods discussed in this paper, in terms of latency and memory efficiencies at different compression ratios?

---

### Official Review · Reviewer_FYE9 · 2024-11-01

**Soundness:** 1
**Presentation:** 1
**Contribution:** 2
**Rating:** 3
**Confidence:** 4

**Summary:**

This paper proposes an automatic way to search for the optimal truncation position when compressing large language models with SVD. The authors first empirically show that the truncation position is highly correlated to the final performance. Based on the observation, they modeled the layer importance based on the correlation and design a way to obtain the optimal configuration. Experimental results demonstrate the effectiveness of the designed searching strategy for the truncation position.

**Strengths:**

1. This paper addresses a good research topic: SVD for LLM compression.SS

2. The paper is well-organized.

**Weaknesses:**

1. **Poor Presentation**  Many aspects starting from Section 2.2 to Section 3.2 are not clarified clearly. Specifically,

    1. The data collected in Figure 1 is also used for modeling the correlation, but why the method only needs to collect 6x40=240 data? why only need to measure compression ratio ranging from 10% to 50%?

    2. Why the author collects the modeling data by only applying the uniform compression ratio and greedy search?

    3. The computed upper-bound is also confusing. On the one hand, it is correlated to the manual configuration Fmin, meaning that setting a larger Fmin could increase the performance? On the other hand, it is correlated to the learned modeling parameter, indicating that changing the pre-defined modeling function from linear one to a more precise unlinear one could also impact the upper-bound. Therefore, it is hard to tell whether reaching this upper-bound is truly the optimal solution.

    4. Since the both the empirical observation in Section 2.2 and the modeling in Section 3.1 are highly data-dependent. What if we change the data distribution for this empirical analysis and modeling?

2. **Overclaim on Compression Speed:** The author claims that the search process of the recent work, ASVD, is slow, however, I found that the designed method still needs to measure the end-to-end perplexity under different compression ratios, which is similar to what has been done in ASVD. Additionally, the proposed method runs learning-based algorithm to model the correlation between truncation position and corresponding perplexity, which is also time-consuming. Given these two situation, it is hard to claim that the proposed automatic searching algorithm is more efficient than prior works.

3. **Missing Experiments:**

    1. Pruning-based compression methods are different from the SVD-based ones, and their compression ratios are not exactly equal to the ratio of parameter reduction in LLM. Therefore, it is not fair to compare these two types of methods under the same compression ratio.

    2. Lack of experimental comparison on generation tasks.

    3. Lack of comparison on quantization-based compression methods.

    4. Lack of analysis on running the methods using data with different distribution.

**Questions:**

See above.

---

### Official Review · Reviewer_Ufkm · 2024-11-02

**Soundness:** 2
**Presentation:** 1
**Contribution:** 2
**Rating:** 3
**Confidence:** 3

**Summary:**

The paper presents AutoTrunc, an automated framework for selecting optimal truncation positions in singular value decomposition (SVD) to compress large language models (LLMs) efficiently. Unlike previous methods that overlook layer importance, AutoTrunc uses a learning-based approach to model each layer’s contribution to overall performance, optimizing truncation to maximize compression while preserving model accuracy. By addressing the truncation selection problem as 0-1 Knapsack Problem with efficient algorithms and dynamically allocating memory based on layer sensitivity, AutoTrunc achieves superior compression results. Experimental evaluations show that AutoTrunc outperforms existing SVD-based methods, reducing perplexity by up to 38.63% on LLaMA-2-13B at a 50% compression ratio, enabling more efficient LLM deployment without retraining.

**Strengths:**

1. **Automated and Importance-Aware Truncation Selection**: AutoTrunc automates the selection of optimal truncation positions using a learning-based approach to model layer importance, focusing on layers critical to performance. This approach streamlines the compression process, maximizing compression efficiency while maintaining accuracy.
2. **Theoretical Soundness**: Built on rigorous NP-hard problem analysis and efficient budget allocation strategies, AutoTrunc is a theoretically grounded method, ensuring reliable performance estimates and compression quality across applications.

**Weaknesses:**

1. **Limited Model Diversity**: The experiments focus solely on the Llama-2 model family, which uses multi-head attention. However, recent open-source LLMs, such as the Llama-3 and Qwen-2/2.5 families, have adopted Group-Query Attention, which might lead to different outcomes in weight compression. This lack of diversity in model structures limits the generalizability of the findings.
2. **Limited Throughput Improvement**: AutoTrunc achieves only modest gains in inference throughput (approximately 1.1x from 0% to 60% compression), whereas other methods, such as SVD-LLM, achieve over 2x speedup at similar compression levels. This limited throughput improvement may reduce AutoTrunc’s impact in applications where throughput is a critical factor.
3. **Lack of Comparison with Quantization Techniques**: The paper does not thoroughly compare AutoTrunc’s performance against other popular compression methods, such as quantization (AWQ, GGUF, GPTQ). Without these comparisons, it is challenging to assess AutoTrunc’s effectiveness, especially in contexts where quantization might offer a better trade-off between compression and performance.

**Questions:**

1. The 0-1 Knapsack Problem is known to be NP-hard primarily because its capacity can be as large as $2^n$. However, in this problem, the capacity is limited. Could this constraint make a brute-force search feasible?
2. Quantization-based model compression techniques, such as W4A16, can reduce model size to 25%. If singular value decomposition (SVD) is combined with quantization, could this approach yield further compression? Given that different decomposition methods offer varying levels of precision, is it possible to identify an SVD approach with high tolerance to quantization?

---

### Note · Authors · 2024-11-21

I have read and agree with the venue's withdrawal policy on behalf of myself and my co-authors.